



# Phosgene distribution derived from MIPAS ESA v8 data: intercomparisons and trends

Paolo Pettinari[1,2], Flavio Barbara[3], Simone Ceccherini[3], Bianca Maria Dinelli[2], Marco Gai[3], Piera Raspollini[3], Luca Sgheri[4], Massimo Valeri[5], Gerald Wetzel[6], Nicola Zoppetti[3], and Marco Ridolfi[1,7]

[1]Dipartimento di Fisica e Astronomia, Università di Bologna, Bologna, Italy
[2]Istituto di Scienze dell'Atmosfera e del Clima, – CNR, Bologna, Italy
[3]Istituto di Fisica Applicata "Nello Carrara" del Consiglio Nazionale delle Ricerche, Via Madonna del Piano 10, 50019 Sesto Fiorentino, Italy
[4]Istituto per le Applicazioni del Calcolo "Picone" – CNR, Firenze, Italy
[5]Serco Italia S.p.A., Frascati (Rome), Italy
[6]Karlsruhe Institute of Technology – IMK-ASF, Karlsruhe, Germany
[7]Istituto Nazionale di Ottica – CNR, Firenze, Italy

**Correspondence:** Paolo Pettinari (p.pettinari@isac.cnr.it), Piera Raspollini (p.raspollini@ifac.cnr.it)

**Abstract.** The Michelson Interferometer for Passive Atmospheric Sounding (MIPAS) measured the middle-infrared limb emission spectrum of the atmosphere from 2002 to 2012 on board ENVISAT, a polar-orbiting satellite. Recently, the European Space Agency (ESA) completed the final reprocessing of MIPAS measurements, using Version 8 of the Level 1 and Level 2 processors, which include more accurate models, processing strategies and auxiliary data. The list of retrieved gases has been

extended, it now includes a number of new species with weak emission features in the MIPAS spectral range. The new retrieved trace species include carbonyl chloride ($COCl_2$), also called *phosgene*. Due to its toxicity, its use has been reduced over the years, however it is still used by chemical industries for several applications. Besides its direct injection in the troposphere, stratospheric phosgene is mainly produced from the photolysis of $CCl_4$, a molecule present in the atmosphere because of human activity. Since phosgene has a long stratospheric lifetime, it must be carefully monitored as it is involved in the ozone

destruction cycles, especially over the winter polar regions.

In this paper we exploit the ESA MIPAS Version 8 data in order to discuss the phosgene distribution, variability and trends in the middle and lower stratosphere and in the upper troposphere.

The zonal averages show that phosgene volume mixing ratio is larger in the stratosphere, with a peak of 40 pptv between 50 and 30 hPa at equatorial latitudes, while at middle and polar latitudes it varies from 10 to 25 pptv. A moderate seasonal

variability is observed in polar regions, mostly between 80 and 50 hPa.

The comparison of MIPAS/ENVISAT $COCl_2$ v.8 profiles with the ones retrieved from MIPAS/balloon and ACE-FTS measurements highlights a negative bias of about 2 pptv, mainly in polar and mid-latitude regions. Part of this bias is attributed to the fact that the ESA Level 2 v.8 processor uses an updated spectroscopic database.

For the trend computation, a fixed pressure grid is used to interpolate the phosgene profiles and, for each pressure level,

VMR monthly averages are computed in pre-defined 10°-wide latitude bins. Then, for each latitudinal bin and pressure level, a regression model has been fitted to the resulting time-series in order to derive the atmospheric trends.



We find that the phosgene trends are different in the two hemispheres. The analysis shows that the stratosphere of the Northern Hemisphere is characterised by a negative trend, of about -7 pptv/decade, while in the Southern Hemisphere phosgene mixing ratios increase with a rate of the order of +4 pptv/decade. In the upper troposphere a positive trend is found in both
hemispheres.

## 1 Introduction

*Phosgene* ($COCl_2$) is a poisonous gas that was used as chemical weapon during the 1st World War (Fitzgerald, 2008). Today, it is still used by chemical industries, in particular in insecticides, herbicides and for pharmaceuticals preparation, even though its use has declined over the years.

The major source of atmospheric phosgene production is the decomposition of chlorocarbon compounds. In particular, the photolisys of carbon tetrachloride ($CCl_4$), together with the reaction between methyl chloroform ($CH_3CCl_3$) and hydroxyl radical (OH) are its main sources in the lower stratosphere, where phosgene peaks. Losses of phosgene in the stratosphere occur via photodissociation, to produce ClO (Fu et al., 2007). However, since phosgene does not react with OH and since it is a weak absorber of ultraviolet radiation, the vertical transport is faster than this process (Kindler et al., 1995). The main loss

mechanisms of $COCl_2$ in the troposphere are the hydrolysis in cloud water and ocean deposition (Toon et al., 2001).

Phosgene is also produced, to a lesser extent, from the atmospheric degradation of the anthropogenic chlorinated Very Short Lived Substances (VSLS), which have a tropospheric lifetime under 6 months. Namely, these are: dichloromethane (methylene chloride, $CH_2Cl_2$), chloroform (trichloromethane, $CHCl_3$), tetrachloroethene (perchloroethylene, $CCl_2CCl_2$, shortened to $C_2Cl_4$), trichloroethene ($C_2HCl_3$) and dichloroethane ($CH_2ClCH_2Cl$), the most abundant of them being $CH_2Cl_2$, (65% of the

whole chlorinated VSLS budget), then $CHCl_3$ (20%), and $CH_2ClCH_2Cl$ (13%). Degradation of these substances to produce $COCl_2$ and HCl may occur either in the troposphere, where the resulting products are then injected in the stratosphere (Product Gas injection, PGI), or in the stratosphere (Source Gas Injection, SGI). Although industrial emissions of all chlorinate VSLS except $CHCl_3$ dominate over natural sources, these compounds are not yet regulated, therefore, the concentration of some of them is increasing in the atmosphere (Engel and Rigby, 2019). Even if the contribution of chlorinated VSLS to the total

tropospheric chlorine (Cl) is approximately only 3% (Engel and Rigby, 2019), their relative contribution to the Cl abundance may increase in the future, when the concentration of substances regulated by the Montreal Protocol will be significantly reduced.

Singh (1976) performed the first in-situ measurement of atmospheric phosgene concentrations. Twelve years later, phosgene data acquired from aircraft revealed a discrepancy between its measurement and the concentrations predicted on the basis of the

$CCl_4$ photochemical breakdown alone (Wilson et al., 1988). Twelve phosgene profiles were recorded by the MklV spectrometer during nine balloon flights between 1992 and 2000, at latitudes near 34°N and 68 °N, using the solar occultation technique (Toon et al., 2001). Recently, a long term record (2004 – 2016) of $COCl_2$ observations was obtained from the measurements of the Atmospheric Chemistry Experiment-Fourier Transform Spectrometer (ACE-FTS) (Harrison et al., 2019).





Long term records are important to study the $COCl_2$ temporal variation that, in turn, provides information on the temporal
variation of the substances responsible for its production, namely $CCl_4$, $CH_3CCl_3$, as well as Cl-containing VSLS. Recently
Harrison et al. (2019) estimated the trend of $COCl_2$ from ACE-FTS measurements in the period 2004-2016 and compared
the results with the TOMCAT/SLIMCAT three dimensional Chemical Transport Model (CTM). A negative $COCl_2$ trend was
found in the stratosphere, and a positive trend in the upper troposphere, compatible with TOMCAT results, when VSLS are
taken into account. The authors also identified a positive bias of 10-20 pptv of the ACE-FTS measurements with respect to the
model data.

Due to the solar occultation technique used by ACE-FTS, its measurements are denser over the polar regions, with relatively
poor coverage in the tropics. For the estimation of global data and trends, the ACE-FTS sparse spatial sampling makes necessary
the averaging of the occultation data for periods of several months, both to reduce the random error down to acceptable levels
and to enhance the density of its coverage. The Michelson Interferometer for Passive Atmospheric Sounding (MIPAS) covers
the spectral region where phosgene signatures are present, therefore its measurements can also be used to retrieve the phosgene
vertical distribution (Valeri et al., 2016). MIPAS measured the middle-infrared limb emission spectrum of the atmosphere
on board the European Space Agency (ESA) ENVIronmental Satellite (ENVISAT), from 2002 to 2012. Being based on the
observation of the infrared limb emission, MIPAS/ENVISAT data are quite dense and uniform both in space and time, providing
a better temporal and spatial coverage than the ACE-FTS measurements. Due to the lower intensity of the measured signal, the
individual MIPAS spectra show a lower signal to noise ratio (SNR) than the ACE-FTS ones. Despite the larger noise error of the
MIPAS profiles retrieved from single scans, Valeri et al. (2016) showed that high vertical resolution, good quality distributions
of $COCl_2$ can be obtained by MIPAS, on a per profile basis. Therefore Valeri et al. (2016) was able to study the seasonality
and the latitudinal distribution of phosgene, based on a restricted set of MIPAS measurements: the data acquired over 2 days
per month in the year 2008.

Recently, ESA completed the final full MIPAS mission re-analysis with Version 8 of both the Level 1 and Level 2 processors.
As compared to earlier versions, these processors include several new features, leading to a significant improvement in the
products accuracy (Raspollini et al., 2021). Among the new features, the list of retrieved atmospheric constituents has been
significantly extended, and, based on the work of Valeri et al. (2016), also phosgene has been included. In this work we analyze
the MIPAS level2-v8 phosgene dataset (Dinelli, 2021) in terms of global distribution, seasonality and trends.

Here, the paper organization is described. In Sect. 2 MIPAS measurements are introduced. In Sect. 3 the phosgene mean
global distribution and seasonality at different latitudes are discussed. In Sect. 4 MIPAS phosgene profiles are compared to
those retrieved from ACE-FTS and MIPAS-balloon (MIPAS-B) measurements. In Sect. 5 the trend estimation strategy is
illustrated and the results are discussed. Finally, in Sect. 6 the analysis conclusions are drawn.

## 2 MIPAS measurements and retrievals

The Fourirer transform spectrometer MIPAS (Fischer et al., 2008), on board the European satellite ENVISAT which lies in
a polar sun-synchronous orbit, acquired atmospheric limb emission spectra from July 2002 to April 2012 in the mid-infrared





region, from 680 to 2410 $cm^{-1}$. In the first period of the MIPAS mission, between July 2002 and March 2004, measurements were acquired, almost continuously, at the full spectral resolution (FR) of 0.025 $cm^{-1}$. This measurement strategy was stopped on 26 March 2004, due to a mechanical problem in the interferometer drive unit. On January 2005, measurements were resumed with a reduced spectral resolution of 0.0625 $cm^{-1}$. Since the new measurement strategy was faster than the FR one, the spatial sampling was increased and the new measurements, acquired after January 2005, belong to the so called *optimized resolution* (OR) period. In addition to the finer horizontal and vertical spatial sampling, the OR measurements have also a reduced Noise Equivalent Spectral Radiance (NESR). Details regarding FR and OR MIPAS measurements can be found in Raspollini et al. (2013, 2021); Dinelli (2021). The important aspect is that MIPAS data, acquired during both mission phases, have a dense coverage over the whole globe, allowing to study the atmospheric composition and its evolution in great detail.

Radiometrically calibrated and geolocated spectra are created by the Level 1b processor (Kleinert et al., 2007) from the MIPAS interferograms. Version 8 of the Level 1b processor includes several improvements as compared to earlier versions (Kleinert et al., 2018). The most relevant for our analysis is the *detector non-linearity correction*. Since photoconductive detectors are subject to ageing, their response slowly decreases with time and becomes more linear as a function of the input signal. If not properly corrected, this effect causes a drift of the instrument response and thus of the retrieved profile values. Starting from Version 7 of the Level 1b processor, time-dependent coefficients, determined from in-flight characterisation , are used to correct for detector non-linearity. In the Level 1b processor version 8, the algorithm used to compute these coefficients has been further refined. Now it accounts for the actual instrument temperature and for the degree of ice contamination. This correction leads to a further reduction of the drift caused by the progressive ageing of the photoconductive detectors. Now the drift error is estimated to be less than 0.5% across the entire MIPAS mission (Kleinert et al., 2018).

The calibrated Level 1b limb emission spectral radiances are further processed by the ESA Level 2 processor version 8, that now coincides with the scientific prototype code named Optimized Retrieval Model (ORM, Ridolfi et al. (2000); Raspollini et al. (2006, 2013)). Full details and an assessment of the improvements implemented in the ORM version 8 (v8) as compared to earlier versions, are provided in Raspollini et al. (2021). Here we only list the main new features of the ORM v.8, relevant to the results presented in this paper.

- Use of the optimal estimation method (Rodgers, 2000) to retrieve the VMR of gases with extremely low signal to noise ratio.

- Modelling of the horizontal variability of the atmosphere using horizontal gradients of temperature, ozone and water vapour, extracted from ECMWF ERA Interim reanalysis at the geolocation and time of each limb scan measurement.

- Use of latitude- and height- dependent cloud-index thresholds to detect and filter limb measurements affected by clouds.

- Use of the updated spectroscopic database HITRAN_mipas_pf4.45 (Raspollini et al., 2021), containing in particular updated data for $COCl_2$ (Tchana et al., 2015) and $HNO_3$ (Perrin et al., 2016), now contained also in HITRAN_2016 database (Gordon et al., 2017). Updated absorption cross-section data for some heavy molecules: CFC$-$11 (Harrison, 2018), CFC$-$113 (Bris et al., 2011) and CFC$-$12, CFC$-$14, $CCl_4$, HCFC$-$22, $ClONO_2$, $HNO_4$ taken from HITRAN 2016 (Gordon et al., 2017).



| FR microwindows ($cm^{-1}$) | OR microwindows ($cm^{-1}$) |
|---|---|
| 843.625 - 846.625 | 843.8125 - 846.8125 |
| 849.200 - 852.200 | 848.0000 - 851.0000 |
| 860.275 - 863.275 | 839.2500 - 842.2500 |
| 838.275 - 840.950 | 859.7500 - 862.7500 |
| / | 854.0000 - 857.0000 |

**Table 1.** Microwindows used for the $COCl_2$ retrievals from FR (left column) and OR (right column) measurements.

As reported in Dinelli (2021), for the retrieval of $COCl_2$, the optimal estimation functionality of the ORM v.8 has been exploited. The used a priori $COCl_2$ profile is a global climatological average (Remedios et al., 2007; Raspollini et al., 2021), independent of both latitude and time. The choice of using a fixed a priori profile to process all measurements guarantees that the observed variability in the retrieved profiles can be solely ascribed to the variability of the measurements. The covariance matrix

representing the a priori error has been built assuming an error made of two components: an absolute, altitude independent component of 1 pptv, and a relative component equal to the 95% of the a priori profile. The off-diagonal elements of this covariance matrix are computed assuming a vertical correlation length of 6 km.

The $COCl_2$ retrieval is performed using optimized spectral intervals (microwindows, MWs) chosen by the MWMAKE algorithm (Dudhia et al., 2002), which selects MWs by minimizing a cost function that takes into account the total retrieval

error achieved when a given test set of MWs is used at specific tangent heights. Therefore, the algorithm also provides an optimized vertical range for the retrieval and the resulting error components of the retrieved profile. The MWs selected for $COCl_2$ retrieval are listed in Table 1. Note that, due to recent updates in the MWMAKE algorithm and in its assumed error figures (Dudhia, 2019), the selected MWs are different from those used in the work of Valeri et al. (2016). The selection process of the new MWs has removed the strong interference with the $CFC-11$ emissions found by Valeri et al. (2016), enabling the

retrieval of $COCl_2$ as a single target. The optimised vertical retrieval range for $COCl_2$ has been estimated to be from 6 to 36 km for the FR measurements and from 9 to 54 km for the OR measurements.

Fig. 1 shows the most relevant error components affecting the phosgene mid-latitude day profiles for the FR (top panel) and OR (bottom panel) MIPAS measurements. In Fig. 1, the error propagating from the measurement noise is labelled as "noise", the error induced by uncertainties in the assumed temperature and pressure profiles are labelled "pt". The errors due to the

uncertainty in the spectrally interfering gases are labelled with the gas name on the plot's labels. The label "hialtx" refers to the error due to assuming the climatological profile shape above the highest point of the retrieval grid, "nonlte" is the error arising from the Local Thermal Equilibrium assumption in the radiative transfer model, "lut" is the error due to using in the radiative transfer cross-section Look-Up-Tables (LUT) compressed with the truncated singular value decomposition method, "gain-a" is the error in the radiometric calibration of the instrument and, finally, "specdb" is the error due to the uncertainties

in the spectroscopic parameters. A more exhaustive list and a more thorough description of the various error components is presented in Dudhia (2019). All the errors presented in Fig. 1, except the "noise" and the "pt" errors, have been calculated



by MWMAKE, by propagating the uncertainties coming from both the instrument and the radiative transfer model onto the retrieved profiles. The "noise" and "pt" errors are a yearly average of the "noise" and "pt" error components, as computed by the ORM v.8 (Raspollini et al., 2021).

The most important contributions to the total error are due to the measurement noise, to the error in tangent pressures and the temperature profile ("pt"), and to the spectroscopic errors ("spcdb"). When the retrieved profiles are averaged, the individual error components combine differently according to their variability from profile to profile. In our analysis, "noise", "pt" and all errors due to the interference with other gases, for example "ocs" or "f11", are considered as random i.e. in the averages they are assumed to decrease inversely to the square root of the number of averaged profiles. Indeed, interference uncertainties depend on the difference between the assumed and real concentrations of the interfering gas, which is expected to change from

scan to scan. On the other hand, spectroscopic ("spcdb"), look up table ("lut") and radiometric calibration ("gain-a") errors do not change from profile to profile, thus they are assumed as systematic and are not damped in the averaging process.

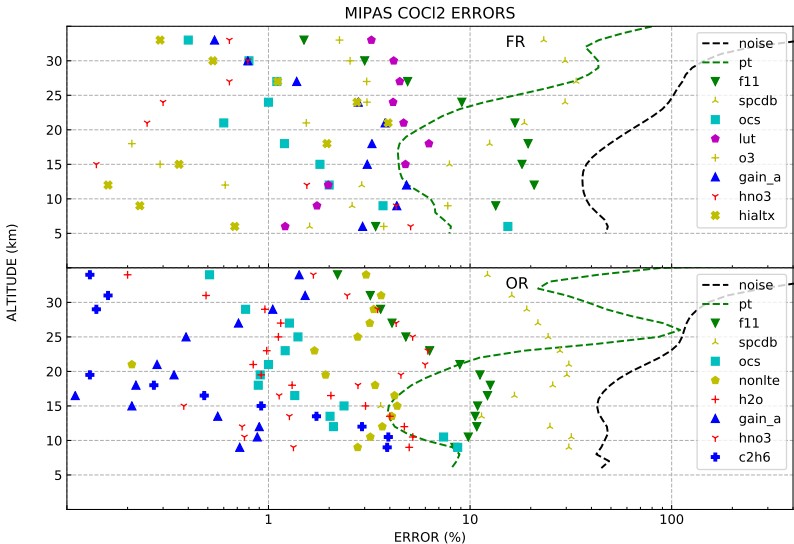

**Figure 1.** Error components affecting the individual retrieved $COCl_2$ profiles in mid-latitude day conditions. The top plot refers to FR measurements, the bottom plot to OR.

The Averaging Kernel (AK) matrix represents a linear approximation of the vertical response function that characterises the measurement chain (Rodgers, 2000; Ceccherini and Ridolfi, 2010). Both the instrument and the retrieval characteristics

contribute to the AK. With the retrieval setup described above, the individual phosgene retrieved profiles show typical AKs (rows of the AK matrix) as in Fig. 2 for FR (left) and OR (right) mission phases. AK peak values approaching unity denote heights where the contribution of the measurements to the retrieval is significantly larger than that of the a priori information. The trace of the AK matrix quantifies the number of Degrees Of Freedom (DOFs) of the retrieval, i.e. the number of independent





pieces of information on the profile extracted from the measurements. As indicated in the plot's key, the number of DOFs of the

individual profiles is around 4. The vertical resolution of the retrieved profiles can be estimated on the basis of the Full Width

at Half Maximum (FWHM) of the AKs, using the formula proposed in Eq. (8) of Ridolfi and Sgheri (2009). According to that

equation, wider AKs correspond to lower vertical resolution. This is confirmed in Fig. 2, showing that the vertical resolution

decreases with altitude. Fig. 2 also shows that the AK diagonal elements become smaller and smaller as altitude increases. For

this reason in our analysis we use $COCl_2$ retrieved profiles only up to 28 km.

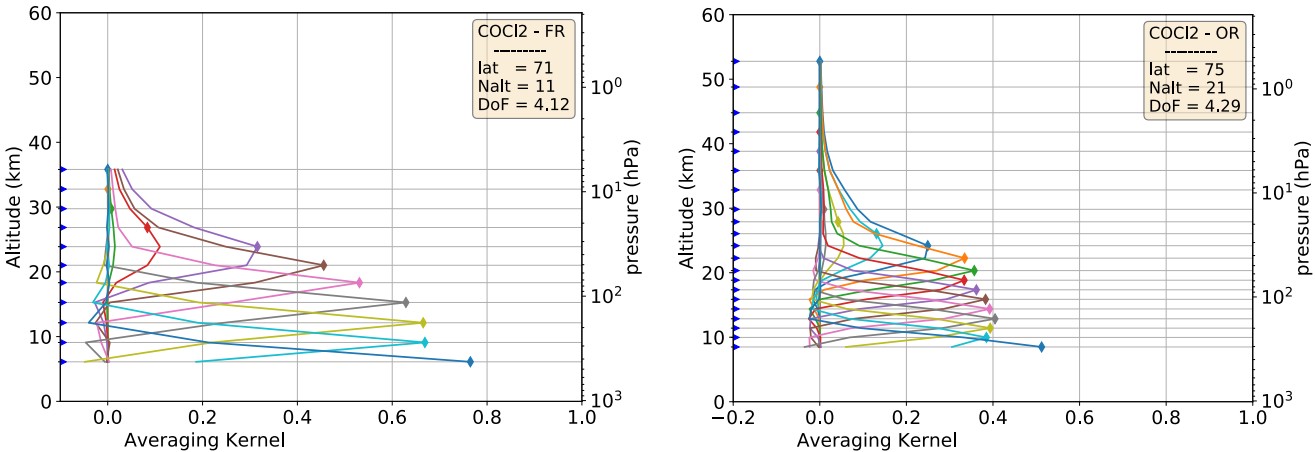

**Figure 2.** Typical AKs of individual $COCl_2$ profiles retrieved from FR (left) and OR (right) MIPAS measurements.

## 170   3   $COCl_2$ average distribution and variability

We interpolated each of the retrieved $COCl_2$ profiles to the SPARC data initiative (Hegglin et al., 2013) pressure grid, con-

sisting of the following 28 levels: 300, 250, 200, 170, 150, 130, 115, 100, 90, 80, 70, 50, 30, 20, 15, 10, 7, 5, 3, 2, 1.5, 1, 0.7,

0.5, 0.3, 0.2, 0.15, 0.1 hPa.Then, we grouped the interpolated profiles into 18 latitude bins, each 10° wide. For each pressure

level and latitude bin, we computed the $COCl_2$ average VMR over the whole MIPAS mission. The left panel of Fig. 3 shows

the global vertical distribution of the $COCl_2$ VMR as a function of latitude. Its distribution is driven by the Brewer-Dobson

circulation and presents the largest VMR values located at the equatorial regions between about 30 and 50 hPa. This behavior

is due to the fact that phosgene is mainly produced by the photolysis of $CCl_4$, thus the peak is located where the sun contri-

bution is maximum and $CCl_4$ is available. The $CCl_4$ global distribution, as obtained by ORM v8 with the same procedure of

phosgene, is shown in the right panel of Fig. 3. We see that the $COCl_2$ peak is located in the lower stratosphere, well above

the maximum of $CCl_4$. This mismatch stems from the fact that at low altitudes, $CCl_4$ photodissociation becomes less efficient,

moreover $COCl_2$ destruction processes, involving hydrolysis, start to become important. In polar regions phosgene VMR does

not exceed the value of 25 pptv because of the low insolation. In troposphere, the VMR is below 15 pptv.





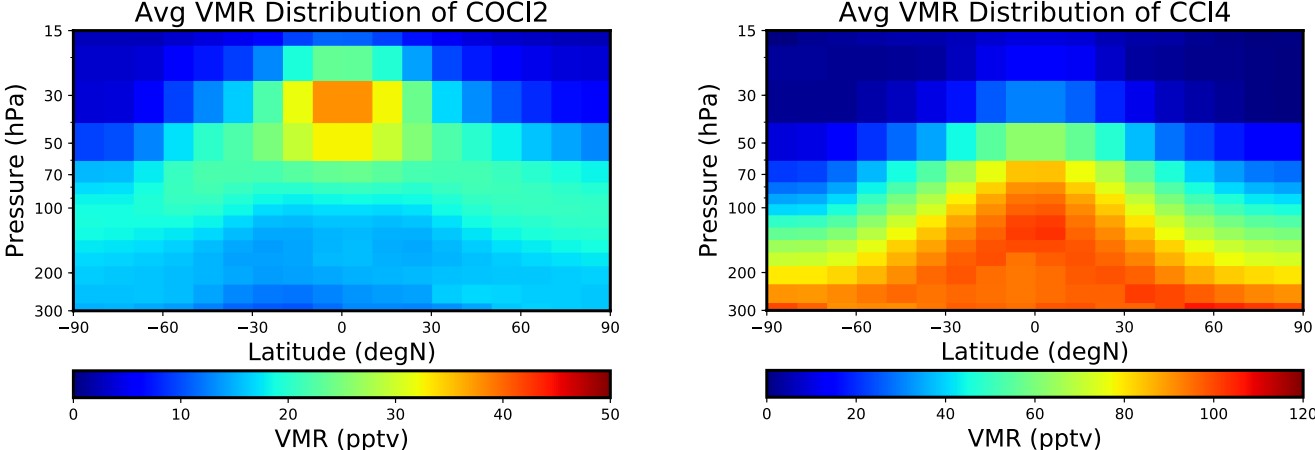

**Figure 3.** COCl₂ (left) and CCl₄ (right) global distributions. Average over the whole MIPAS mission period (2002–2012).

The four panels of Fig. 4 illustrate the seasonal and latitudinal variability of the phosgene VMR, at four different pressure levels. In general, the seasonal variability is more pronounced in polar regions that are subject to both the largest insolation variability during the year and the presence of a strong vortex during wintertime. At high altitudes (15 hPa), as shown in the top left panel of Fig. 4, phosgene concentrations are small and rather constant in time, both at the equator, because of the stable insolation, and at the poles, because at this pressure level $CCl_4$ concentration is very small. Moving downward, down to 50 hPa, the $COCl_2$ VMR increases almost everywhere except at high latitudes during the polar nights. The periodic change in the insolation at the poles originates a $COCl_2$ variability in polar regions, with an alternation between summer peaks and winter deficiencies. The seasonal oscillation becomes less evident at 100 hPa, as shown in the bottom right panel of Fig. 4. This is due to the smaller efficiency of the $CCl_4$ photodissociation.

## 4    Comparisons to balloon and other satellite measurements

To the best of our knowledge, the only available $COCl_2$ measurements that cover the time period of the ENVISAT mission are those of the Atmospheric Chemistry Experiment - Fourier Transform Spectrometer (ACE-FTS) and those of the balloon version of the MIPAS instrument (MIPAS-B). In this section we show the results of the comparison of our products with these measurements.

### 4.1    Comparison to ACE-FTS

ACE-FTS is a Fourier transform spectrometer onboard the SCISAT-1 satellite, launched in August 2003. Starting from February 2004, ACE-FTS has been acquiring solar occultation measurements in the middle infrared (Bernath et al., 2005; Bernath, 2017). From these spectral absorption measurements it is possible to infer the vertical distribution of many atmospheric con-





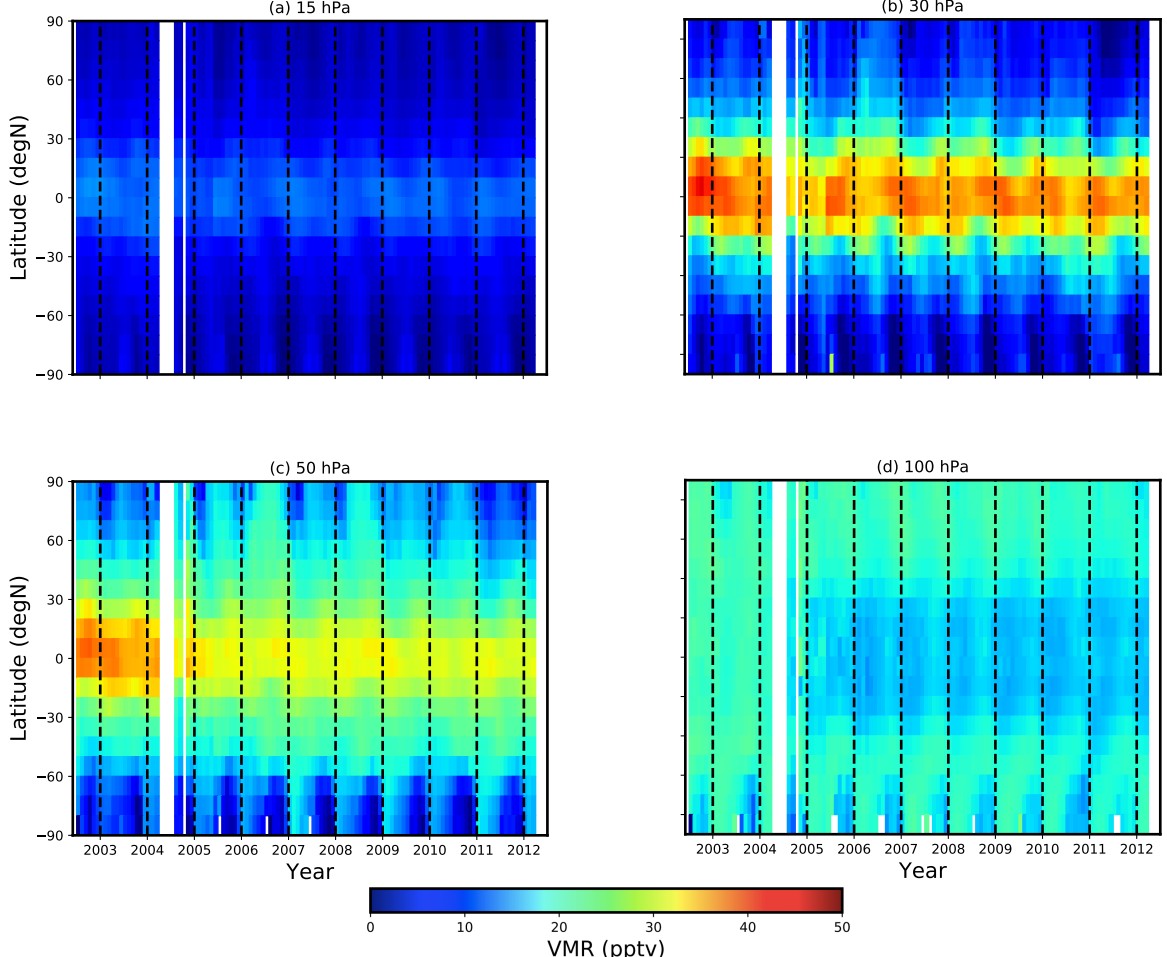

**Figure 4.** Time series of monthly mean $COCl_2$ VMR as a function of latitude (vertical scale). The four panels refer to different pressure levels: 15 hPa (top left), 30 hPa (top right), 50 hPa (bottom left) and 100 hPa (bottom right). The white areas denote periods of missing measurements. Labels in the x-axis indicate the beginning of the years.

stituents (Hase et al., 2010). For this intercomparison exercise, we use ACE-FTS v3.5 $COCl_2$ products (Koo et al., 2017), retrieved using the spectroscopic database described in Brown et al. (1996) for $COCl_2$.

Due to the solar occultation technique, ACE-FTS measurements are quite dense near the poles and sparse in the equatorial regions. For this reason, to have a distribution of co-located measurements as homogeneous as possible all over the globe, latitude-dependent matching criteria have to be used. After some tuning, we adopted the matching criteria described in Table 2. We verified that using more stringent criteria, the conclusions of the intercomparison to ACE measurements won't change.





| Lat band (deg) | Max. allowed distance (km) | Max. time mismatch (hours) |
|---|---|---|
| 90-60 N/S (poles) | 300 | 6 |
| 60-30 N/S (mid-latitudes) | 600 | 8 |
| 30-0 N/S (tropics) | 900 | 10 |

**Table 2.** Latitude-dependent matching criteria used for the intercomparisons with ACE-FTS measurements.

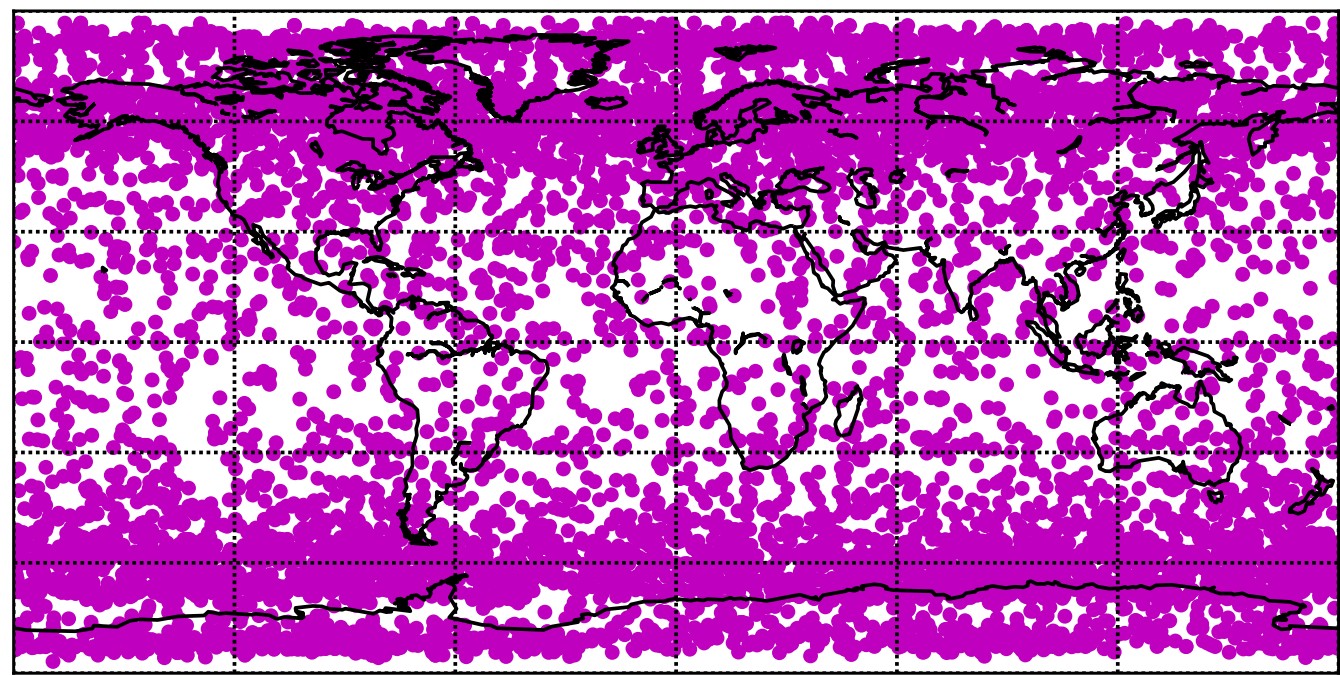

**Figure 5.** Distribution of MIPAS/ESA v.8 and ACE-FTS v.3.5 matching measurements.

This choice provides matching measurements distributed as shown in Fig. 5, allowing to perform a statistically significant intercomparison in all the relevant latitude bands. Note, however, that the number of coincidences in the tropical region is still significantly smaller than at polar- and mid- latitudes.

First, we interpolated the matching $COCl_2$ profiles to a fixed altitude grid made of the following 14 levels: 9, 10, 11, 12.5, 14, 15.5, 17, 18.5, 20, 21.5, 23, 24.5, 26 and 27.5 km. We then computed the median and the mean of the profile differences for the three latitudinal bands indicated in Table 2. The results are illustrated in Fig. 6.

Fig. 6 shows the average (blue) and median (green) differences between MIPAS and ACE-FTS $COCl_2$ profiles. The blue error bars represent the total random error of the average differences, while the red curves represent the $\pm$ combined systematic error bounds. The total systematic error of the average differences was computed as the quadratic summation of MIPAS and ACE-FTS systematic errors. For MIPAS systematic errors, we assumed only the contributions from "spcdb", "lut" and "gain-



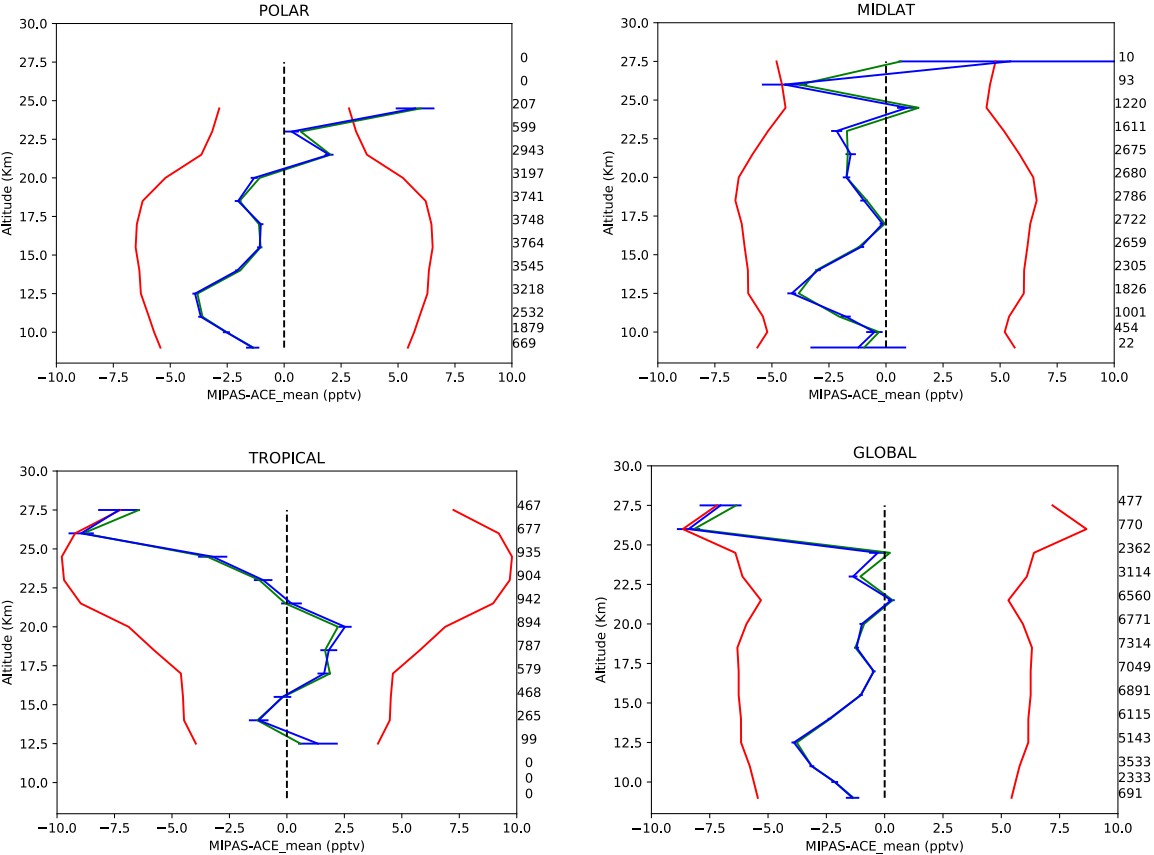

**Figure 6.** Average differences (blue) between MIPAS and ACE-FTS $COCl_2$ profiles for several latitude bands (see plot's key). The median differences are shown in green. Blue bars and red lines represent the random and systematic combined error components, respectively. The number of matching measurements is indicated on the right hand axis.

a" as described in Sect. 2. The assumed systematic error on ACE-FTS profiles is 30%, as suggested in Fu et al. (2007). The combined random error is estimated as the standard deviation of the profile differences divided by the square root of the number of matching measurements.

We see that MIPAS systematically underestimates ACE-FTS in polar- and mid- latitudes at almost all altitudes. On aver-age, the negative bias is generally smaller than 2 pptv, with the exception of the altitudes around the upper troposphere-lower stratosphere, where the negative difference peaks to almost 4 pptv. On the other hand, the equatorial mean differences oscillate around zero, changing from positive values smaller than 2 pptv at about 18 km, to negative values, decreasing with altitude, down to about -7 pptv at 26 km. Note that the observed systematic differences are mostly comparable to the estimated sys-

tematic errors, therefore the MIPAS phosgene profiles have a satisfactory agreement with the ACE-FTS ones. It is important to note that the discrepancy with ACE-FTS at high altitudes is not observed, in the comparison between MIPAS-ENVISAT



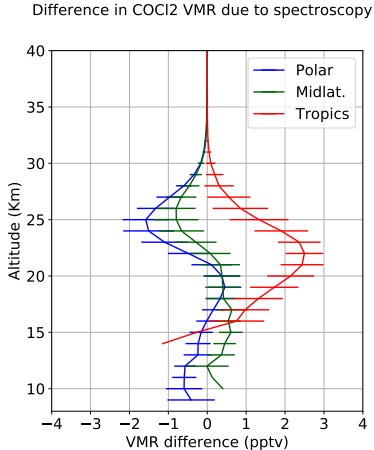

**Figure 7.** Average differences between COCl$_2$ retrieved by ORM V8, using the spectroscopic database described in Gordon et al. 2017, and in Brown et al. 1996. The average was computed using the scans of one orbit of OR measurements, grouped into 3 latitude bands as described in the label of the figure. The error bars are the errors of the average.

(MIPAS-E) and MIPAS-balloon (MIPAS-B) (see Fig. 8 in the next Section). Part of the systematic differences between MIPAS-E and ACE-FTS can be ascribed to the different spectroscopic databases used in the analysis. Indeed, the analysis of ACE-FTS was performed with COCl$_2$ spectroscopic database from Brown et al. (1996), while the MIPAS-E analysis uses the newer

spectroscopic database (Tchana et al., 2015; Gordon et al., 2017). Fig. 7 shows the average difference between COCl$_2$ profiles retrieved with ORM V8 using the two different spectroscopic databases. Some features of these differences are similar to those observed in the intercomparison of Fig. 6. Specifically, we refer to the negative bias in polar regions, and to the shape of the difference profile in the equatorial regions.

Finally, the negative bias that MIPAS profiles have with respect to ACE-FTS ones should not be exclusively regarded as a

deficiency in MIPAS data. Indeed, while comparing the upper tropospheric ACE-FTS COCl$_2$ to the chemical transport model TOMCAT (Monks et al., 2017), Harrison et al. (2019) found a positive bias of $10 - 20$ pptv between measurements and model, while the results of the present intercomparison hint to a slightly smaller bias between MIPAS and TOMCAT concentrations. However, the inconsistency is still not completely solved.

### 4.2 Comparison to MIPAS-balloon

The MIPAS-E phosgene profiles have been compared also to MIPAS-B (Wetzel et al., 2018) measurements. Since the balloon-borne sounder MIPAS-B has a number of instrumental characteristics similar to MIPAS/E, like the spectral coverage (750–2500 cm$^{-1}$) and the spectral resolution (0.0345 cm$^{-1}$), it can be considered as its precursor. However, the radiometric (NESR) and pointing performances of MIPAS-B are superior. In particular, the error of the tangent altitude is reduced to 90 m ($3\sigma$), thanks to the new line of sight stabilization. Also the MIPAS-B NESR is improved by computing the average of several spectra





| Location | Date | Distance (km) | Time difference (min) |
|---|---|---|---|
| Kiruna (68 N) | 20 Mar 2003 | 16/546 | 14/15 |
| | 03 Jul 2003 | Trajectories only | |
| | 11 Mar 2009 | 187/248 | 5/6 |
| | 24 Jan 2010 | 109/302 | 5/6 |
| | 31 Mar 2011 | Trajectories only | |
| Aire-sur-l'Adour (44 N) | 24 Sep 2002 | 21/588/410/146 | 12/13/15/16 |
| Teresina (5 S) | 14 Jun 2005 | 109/497/184/338 | 228/229/268/269 |
| | 06 Jun 2008 | 224/284/600/194 | 157/158/169/170 |

**Table 3.** Overview of MIPAS-B flights used for intercomparison with MIPAS/ENVISAT

acquired at the same elevation angle. Generally, every measurement of a MIPAS-B scan is recorded every 1.5 km of vertical tangent altitude. The least squares fitting algorithm used to retrieve all the species, on a 1 km grid, is based on analytical derivative spectra computed by the Karlsruhe Optimized and Precise Radiative transfer Algorithm (KOPRA) (Höpfner et al., 1998; Stiller et al., 2002). A regularization approach constrains the difference between the first derivative of the retrieved and an a-priori profile in order to avoid retrieval instabilities due to the vertical oversampling. The MIPAS-B phosgene retrieval

is performed in the spectral interval between 838.8 and 860.0 $cm^{-1}$. Spectroscopic parameters from both the MIPAS-E dedicated spectroscopic database (Raspollini et al., 2013; Perrin et al., 2016) and the HITRAN 2008 database (Rothman et al., 2009) are used to simulate the infrared emission spectra. For the $COCl_2$ spectroscopic database, data from Brown et al. (1996) are used. All the details regarding the MIPAS-B error estimation and data analysis can be found in Wetzel et al. (2018). The intercomparison with MIPAS-E is performed using the MIPAS-B flights listed in Table 3. For the intercomparison, both direct

matches and trajectory matches are considered. Direct matches occur when both instruments sound the same air-mass simultaneously. While, in the case of trajectory matches, forward and backward trajectories were calculated by the Free University of Berlin (Naujokat and Grunow, 2003) from the balloon measurement geolocation, in order to look for air-masses observed by MIPAS-E.

The coincidence criterion, adopted for the identification of both direct and trajectory matches, was of 1 hour and 500 km.

For a direct comparison to the MIPAS-B data, the MIPAS-E VMR and temperature profiles were interpolated to the altitudes relative to the trajectory matches.

The results in Fig. 8 indicate that MIPAS-E profiles result generally underestimated for arctic and mid-latitude atmospheric conditions with respect to MIPAS-B. In the tropics, the differences oscillate around zero, MIPAS-E being overestimated in the lower stratosphere-upper troposphere, and underestimated below and above.





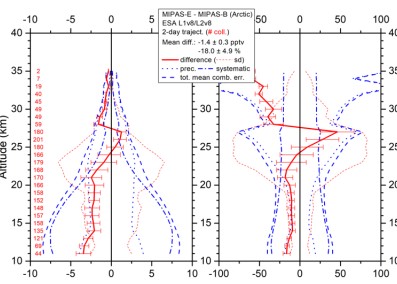 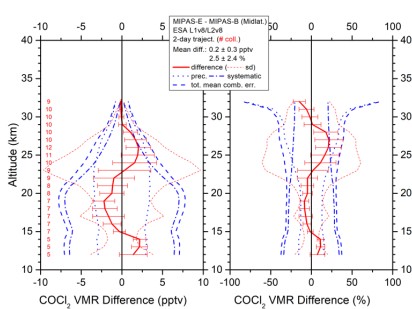 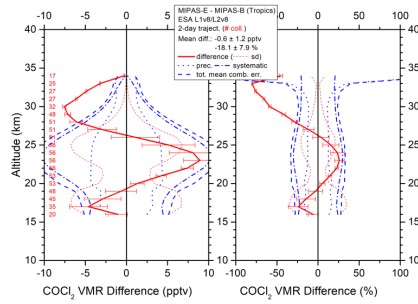

**Figure 8.** Average differences between MIPAS-E and MIPAS-B (Wetzel et al., 2018). The plots show the absolute (left) and relative (right) VMR mean differences between MIPAS-E and MIPAS-B (red solid line) for the match collocations found (red numbers). The standard deviation of the mean and standard deviation of the differences are represented by the error bars and the red dotted lines respectively. The other curves displayed are: systematic (blue dotted, dashed lines), precision (blue dotted lines) and total combined errors (blue dashed lines) obtained adding the errors in quadrature.

Despite some systematic differences, the consistency between the two MIPAS instruments is good, indeed the mean differences, with their error bars, are smaller than the estimated systematic errors, almost everywhere. Some exceptions occur at high altitudes.

The behaviour of the differences between MIPAS-E and MIPAS-B is still similar to the one obtained in the comparison between MIPAS-E and ACE-FTS presented in Fig. 6, although the amplitude of the differences and the location of the maxima are in some cases different. As already mentioned, the retrievals from both ACE-FTS and MIPAS-B are performed with the $COCl_2$ spectroscopic database from Brown et al. (1996), while MIPAS-E retrievals use data from Tchana et al. (2015). Therefore, we can safely say that part of the differences between MIPAS-E and MIPAS-B, are due to the different spectroscopic data used.

## 5 Phosgene trends

To compute $COCl_2$ trends we use the algorithm developed by Valeri et al. (2017). For each pressure level and latitude bin mentioned in Sect. 3, a model $VMR(t)$ is adapted, with the least squares approach, to the time series of the monthly $COCl_2$ averages. As explained in Valeri et al. (2017), the model includes two different offsets for the two mission phases (FR and OR), a common linear term (the trend we are interested in), QBO- and solar flux- related terms and the characteristic atmospheric periodicities of 24, 18, 12, 9, 8, 6, 4, 3 months.

Fig. 9 illustrates the behaviour of the fit of the monthly averages for two specific pressure levels and latitude bins. The left plot refers to the latitude bin centred at 15°N and the pressure level of 100 hPa, the right plot refers to 25°S and 115 hPa. Blue dots are the monthly mean VMRs with their error bars, computed as the standard deviation of the mean. The $VMR(t)$ model and the linear term are represented by the red curve and the green lines respectively, with two different offsets for the FR and the OR mission phases. The slope of the green lines represents the linear trend we estimate. The black lines plotted in the bottom





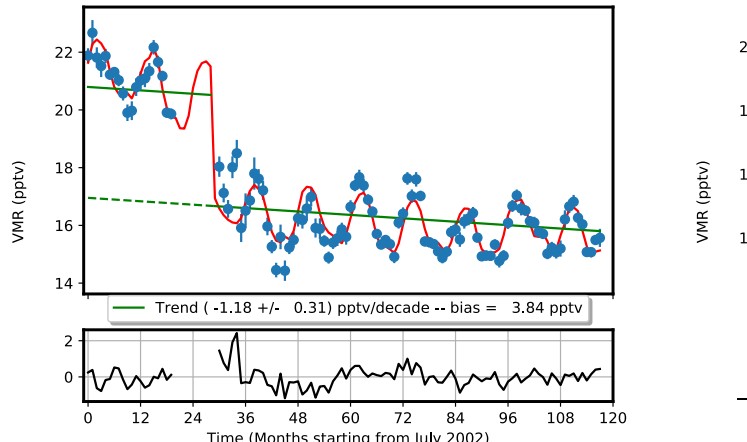 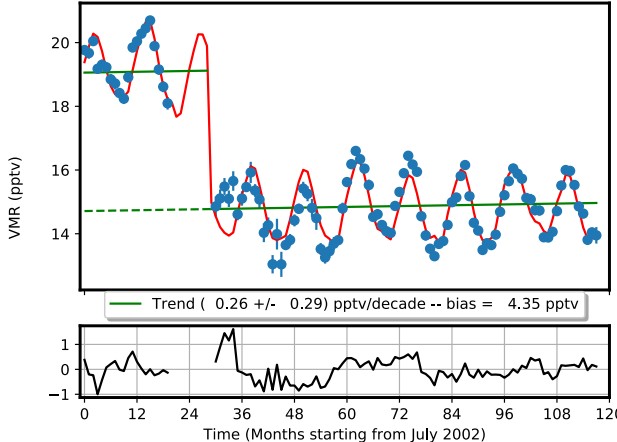

**Figure 9.** Blue dots are the monthly average VMRs with their error bars. The $\text{VMR}(t)$ model and the linear term are represented by the red curve and the green line respectively, with two different intercepts, for the FR and the OR mission phases. The trend is the slope of the green lines. The left plots refer to the latitude bin centred at $15°$N at the pressure level of $100\,\text{hPa}$, the right plots refer to $25°$S and $115\,\text{hPa}$. The black lines in the bottom plots show the residuals of the fits.

panels of Fig. 9 represent the residuals of the fit, computed as the observed monthly means minus the model $\text{VMR}(t)$ values. Note that the residuals oscillate around zero, except in the first half of $2005$ (between $24^{th}$ and $36^{th}$ month), which corresponds to the transition period between FR and OR mission phases. The two plots show that the same model $\text{VMR}(t)$ used in Valeri et al. (2017) to fit the time series of the $CCl_4$ monthly means is able to properly capture also the $COCl_2$ variability. In the fitting procedure we associate to each monthly mean the standard error of the mean. This error accounts for both the variable error 290 components (see Sect. 2) of the individual measurements that contribute to the average, and the temporal and spatial variability of the atmosphere within the considered month and latitude band. Systematic error components such as spectroscopic errors and errors due to lookup tables compression ("lut") provide a constant contribution during the entire mission, thus they do not affect the trend estimate. A different consideration applies to the radiometric calibration error ("gain-a") which, due to the detector ageing, may drift during the mission and, thus, affects the trend estimate. According to the most recent assessment for 295 the Level 1b products v.8 (see Sect. 2 and Kleinert et al. (2007)), however, the drift of the calibration error has been estimated to be less than 0.5 % across the whole MIPAS mission. Since the $COCl_2$ error due to the uncertainty in radiometric calibration ("gain-a", see Fig. 1)) on its own is generally smaller than 5 %, its drift can be considered to have a negligible impact on the estimated trends, as compared to the random error.

In summary, we compute the trend error by simply propagating the monthly mean standard error onto the derived trends. To 300 account for the quality of the retrieval, we multiply this number by the square root of the normalised chi-square of the trend fit (i.e. the normalised error-weighted L2-norm of the residuals of the fit).





Fig. 10 shows the $COCl_2$ absolute trends (a) and their errors (b). Percentage trends (c) and errors (d), computed as the ratio between their absolute values and the average VMR in the corresponding grid cell, are shown in the same figure. In Fig. 11 we show the inverse of the trends relative error. This is a quantifier for the statistical significance of the trends themselves.

Whenever the computed trend is smaller or of the same order of its error, the trend has little statistical significance and, actually, could be either positive or negative. Values of the inverse relative error greater than 2 or 3 indicate a statistically significant trend. Among the determined trends, the one with the largest statistical significance are located in the northern hemisphere.



**Figure 10.** Absolute (a) and percentage (c) $COCl_2$ trends for each latitude and pressure bin. Absolute and percentage trend errors are reported in panels (b) and (d) respectively. Dashed areas represent pressure levels and latitude bins with too few data.

From Fig. 10 we see that negative trends characterize the stratosphere in the northern hemisphere with values down to $-7$ pptv/decade at 30 hPa. In the southern hemisphere the trend values range from negative (about $-2$ pptv/decade), for

pressures between 200 and 70 hPa, to positive (about 4 pptv/decade), above 50 hPa. At high altitudes there is an evident

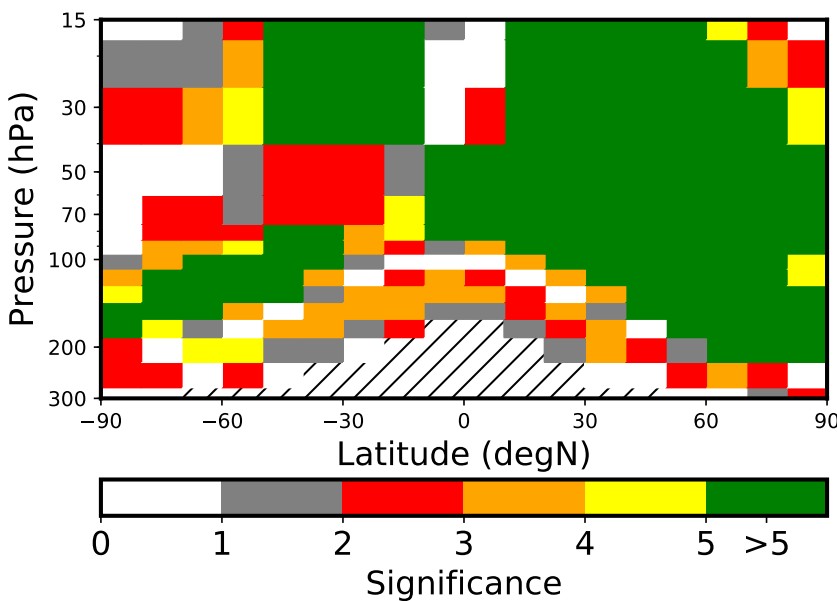

**Figure 11.** Statistical significance (inverse relative error) of the determined trends. Dashed areas represent pressure levels and latitude bins with too few data.

asymmetry between the northern and southern hemispheres. In the upper troposphere we find a positive trend in both hemispheres, even if its statistical significance (see Fig.11) is smaller than in the stratosphere.

A similar analysis of $COCl_2$ trends resolved in latitude and altitude has recently been published by Harrison et al. (2019). In that work, phosgene trends based on ACE-FTS measurements and on the TOMCAT/SLIMCAT model (Chipperfield, 2006)
were derived for the period from January 2004 to December 2016. The trends from ACE-FTS measurements were computed from monthly percentage anomalies in $COCl_2$ zonal means. The authors did not include additional terms such as the annual cycle and its harmonics, claiming that they cause no additional improvements in the regression. The map of the trends versus pressure and latitude derived from MIPAS/ESA v.8 products is consistent with the results from both ACE-FTS measurements and TOMCAT simulations, i.e. the negative trend in the lower stratosphere in both hemispheres, the trend asymmetry in the two
hemispheres in the middle stratosphere, and the positive trend in the upper troposphere. In particular the latter, attributed by Harrison et al. (2019) to the increase of Cl-containing VSLS, confirms the importance of $COCl_2$ as a marker for Cl-containing VSLS.

In their paper, Harrison et al. (2019) also computed the overall VMR-weighted trend from both ACE measurements and TOMCAT simulations. This parameter is calculated as the average of the trends at all latitude and altitude bins, weighted
with the mean VMR in each bin. The results of Harrison et al. (2019) are: $(-0.06 \pm 0.03)$ pptv/year for the trend from ACE



data and $(-0.209 \pm 0.005)$ pptv/year for the trend from the TOMCAT model. To compare more quantitatively MIPAS and ACE-FTS trends we also computed from our MIPAS trends an overall VMR-weighted trend with the same procedure. We get $(-0.15 \pm 0.02)$ pptv/year, where the error has been estimated by propagating the errors of each individual trend and monthly mean VMR. Although ACE-FTS and TOMCAT overall trends are not self-consistent, MIPAS result falls within these two values, at a distance of 2.5 to $3\sigma$ from them.

Concerning the trend asymmetry in the upper stratosphere, a similar feature has been also observed for $CFC-11, CFC-12$ (Kellmann et al., 2012), and for $CCl_4$ (Valeri et al., 2017) trends. According to some recent researches (Harrison et al., 2019; Mahieu et al., 2014; Ploeger et al., 2015), the stratospheric circulation variability is able to affect the stratospheric trends of several halogen source gases and the derived compounds, i.e. HF and HCl. Such results give a partial explanation as to why the stratospheric trend does not always resamble the tropospheric one with a time lag.

## 6   Conclusions

The production of atmospheric phosgene is mainly due to the chlorocarbon compounds decomposition, primarily $CCl_4$ and, secondarily, $CH_3CCl_3$ and VSLS. In this study, the global vertical distribution of upper tropospheric and lower and middle stratospheric $COCl_2$, retrieved by the ESA Level 2 V8 processor (ORM v8) from the full set of MIPAS/ENVISAT measurements in the decade 2002-2012, is reported. We find peak values of phosgene of the order of $\approx 40$ pptv in the tropical lower stratosphere. This is the latitude and altitude where both sun insolation and $CCl_4$ VMR (a species of anthropogenic origin and whose concentration decreases with altitude) are high enough to trigger $COCl_2$ production. Minimum values, approaching 0 pptv, are found between 15 and 30 hPa over polar regions, where both insolation and $CCl_4$ VMR are scarce.

Intercomparisons of MIPAS phosgene profiles with ACE-FTS and MIPAS-B measurements reveal a slight negative bias of MIPAS at polar- and mid- latitudes, of the order of 2 pptv. Analyzing both MIPAS/balloon and ACE intercomparisons, we can say that part of the bias is due to the fact that the retrieval of MIPAS/ENVISAT measurements uses the most recent $COCl_2$ spectroscopic database (Tchana et al., 2015; Gordon et al., 2017).

For the computation of the trends, we interpolated the MIPAS phosgene retrieved profiles at fixed pressure levels and grouped them into $10°$-latitude bins. For each pressure level and latitude bin, we calculated monthly averages for the whole period of the MIPAS mission, from July 2002 to April 2012. By fitting a model to the monthly averages, we determined the $COCl_2$ trends. Negative trends characterise the lowest and middle stratosphere of the northern hemisphere with values down to $-7$ pptv/decade at 30 hPa. In the southern hemisphere the trend values range from negative (about $-2$ pptv/decade), for pressures between 200 and 70 hPa, to positive (about 4 pptv/decade), over 70 hPa. In the upper troposphere we find positive trends at all latitudes in both hemispheres. However, such values are never larger than 1 or 2 times their error bars, therefore their statistical significance is lower than in the stratosphere. A similar tropospheric trend distribution has been found by Harrison et al. (2019) using ACE-FTS measurements. This behavior was attributed to the increase of Cl-containing VSLS, so this result confirms the importance of $COCl_2$ as a marker for Cl-containing VSLS.



As stratospheric $COCl_2$ is primarily generated by the photolysis of $CCl_4$, the stratospheric $COCl_2$ trend is similar to the one of $CCl_4$, which is negative almost everywhere, except in the southern hemisphere, over 50 hPa, where the trend is positive in some regions (Valeri et al., 2017). However, some differences exist. Positive trend in the Southern hemisphere is more evident for $COCl_2$ where peaks of more than 60%/decade are present, while $CCl_4$ does not increase more than 30 %/decade and in a more limited region. Indeed, despite the photolysis of $CCl_4$ is the main stratospheric source of phosgene, other chlorocarbon compounds, like VSLSs, whose stratospheric concentration is increasing (Hossaini et al., 2019), can play an important role. MIPAS $COCl_2$ trends behavior is similar to that found by Harrison et al. (2019) from ACE-FTS results, both the signs and the asymmetry between the two hemispheres are in good agreement. Regarding the overall VMR-weighted trend values, our result of $(-0.15 \pm 0.02)$ pptv/year falls right in between the value inferred from ACE $(-0.06 \pm 0.03)$ pptv/year and the TOMCAT model estimate of $(-0.209 \pm 0.005)$ pptv/year (Harrison et al., 2019).

## 7   Data availability

MIPAS ESA Level 2 products version 8 can be obtained via https://earth.esa.int/eogateway/ (registration required). Trend values are available upon request to the authors.

ACE-FTS level 2 products version v3.5 can be obtained via: http://www.ace.uwaterloo.ca/data.php (upon request).

*Author contributions.*  PR, MR, LS, MG, BMD, FB and SC are the main authors of the MIPAS/ESA ORM v.8 processor. NZ computed from the MIPAS Level 2 v.8 products the monthly means that were used to derive the trends and the global distribution maps. MV and MR designed and implemented the algorithm to derive trends. PP computed trends and compared MIPAS $COCl_2$ profiles to ACE-FTS. Comparisons to MIPAS-B measurements were done by GW. PP, PR and BMD discussed the results. The paper was first drafted by PP, then carefully reviewed by all co-authors. BMD and MR procured funding for the research contract of PP.

*Competing interests.*  The authors declare that they have no conflict of interest.

*Acknowledgements.*  The Atmospheric Chemistry Experiment (ACE), also known as SCISAT, is a Canadian-led mission mainly supported by the Canadian Space Agency.

The MIPAS $COCl_2$ VMR dataset was generated as part of the ESA Level 2 V8 full mission reprocessing of MIPAS measurements performed under ESA contract no. 4000112093/14/I-LG. We thank the MIPAS Quality Working Group for the work done to improve the Level 1 and the Level 2 processors.





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
