# Peer review of "Phosgene distribution derived from MIPAS ESA v8 data: intercomparisons and trends"

_Atmospheric Measurement Techniques, 2021_

## Author Comment (AC1)

General Comments

In this new study, Pettinari et al. discuss the global distribution and trends of phosgene (COCl2) measurements by Envisat MIPAS during the years 2002-2012. A comparison of the MIPAS measurements with ACE-FTS and MIPAS balloon measurements is presented. A 10-year trend analysis is shown, and the phosgene trends found in the MIPAS data are related to different contributing factors, in particular to the distribution and trends of CCl4.

Overall, this is an interesting and carefully conducted study, I think. The manuscript is concise and mostly clear. A few minor suggestions are listed below. In particular, it would be good to add some discussion on how the different vertical resolution of the MIPAS FR and OR modes, the ACE-FTS, and the balloon data affects the results shown here, I think. Once the comments are addressed, I would recommend the paper for publication in Atmospheric Measurement Techniques.

ANSWER: Thank you very much. we are sure that your suggestions will improve this paper.

Specific Comments

l19-25: In the abstract, it would be nice to add a sentence explaining the phosgene trends observed by MIPAS, i.e., refer to the trends of CCl4.

ANSWER: We added two sentences at the end of the abstract in the revised paper.

l94-95: You might add a sentence saying how many vertical profiles are measured each day to provide a number for the "dense coverage".

ANSWER: We added a sentence with this information.

l135-136: This statement suggests the OR mode retrieval works much better and higher up than the FR mode retrieval. Is it really meaningful to say the phosgene retrieval works up to 54 km, considering the averaging kernels shown in Fig. 2 indicate a reasonable upper limit of about 25-30 km?

ANSWER: In this sentence we are just mentioning the retrieval range used for the FR and OR periods. Of course, the retrieval produces independent information only below a certain height. For this reason, this study has been performed using only data inside the so-called COCl2 useful range, which is below 28 km. To explain this, we added a sentence citing the article where the COCl2 useful range is defined.

l167-168: Can you please provide the actual numbers for the vertical resolution of the FR and OR phosgene retrievals?

ANSWER: Yes, the vertical resolution of FR (OR) period is about 5 km (3.5 km) below the altitude of 17 km. Then, it starts to get worse reaching 10 km at the altitude of 25 km. We added a sentence with this information in the revised paper.

Fig. 1: The OR retrievals shows a peak in "pt" errors at 25 km. What is causing this?

[Figure]

Figure A1: pT error propagation matrices for COCl$_2$ VMR retrieved from FR (left) and OR (right) measurements.

The green dashed lines in Figure 1 of the paper represent the profiles of COCl$_2$ VMR error caused by the propagation of the pT random error components. Those curves show a pronounced peak located around 27 km in the OR case, and an increase above $\approx$28.5 km in the FR case. Figure A1 shows the so called pT error propagation matrices for the COCl$_2$ VMR retrieved from FR (left) and OR (right) measurements. For each VMR parameter indexed in the horizontal axis, the color scale indicates the percentage VMR change obtained by applying a perturbation (1K variation in temperature or 1% variation in pressure) to the pT-retrieval vector element indicated in the vertical axis. The first half of the PT-retrieval vector elements refer to the tangent pressures, the second half to the temperature profile grid points. Pressure and temperature parameters are indexed starting from the top of the atmosphere. The red arrows in Figure A1 indicate the VMR parameter indices that roughly correspond to the heights of the peak (OR case, VMR parameter index #10) and of the increase (FR case, VMR parameter index #3) of the pT-errors reported in Figure 1 of the paper. Figure A1 suggests that the pronounced peak of the pT induced error observed in the OR case is caused by an increased sensitivity to pT variations of the VMR at these altitudes. Both the different microwindows used (see table 1 of the paper) and the finer limb scan pattern implemented in the OR mission can actually generate the increased sensitivity observed.

While describing the contents of Figure 1, in the revised version of the paper we added a comment on this regard.

Fig. 2: Can you please add a curve showing the integral of the averaging kernels so that it is more easy to see at which height range the retrieval results are mostly determined by information from the measurements rather than a priori data?

ANSWER:

To highlight the contribution of the actual measurements to the individual retrieved parameters we prefer to use the parameter-specific information gain $q_j$ (first introduced in Dinelli et al. : MIPAS2D database ..., Atmos. Meas. Tech., 3, 355 - 374, 2010 ) defined as:

$q_j = - \frac{1}{2} \log_2 ( S_{x\,jj} / S_{a\,jj} )$

where $S_{x\,jj}$ and $S_{a\,jj}$ denote the $j$-th diagonal elements of the retrieval- and a-priori- error covariance matrices, respectively. If the measurements do not contribute to determine the $j$-th retrieval parameter we get $S_{x\,jj} = S_{a\,jj}$ , thus $q_j = 0$. On the other hand, if the actual measurements contribute to determine the $j$-th parameter, then we get $S_{x\,jj} < S_{a\,jj}$ , thus $q_j > 0$. For example, if $S_{x\,jj} = S_{a\,jj} / 4$ (the measurements are able to halve the a-priori uncertainty), then we get $q_j = 1$, i.e. we gain 1-bit of information.

The plots of figure 2 of the revised paper now include also the curves of $q_j = q(z_j)$. The text describing figure 2 has been modified accordingly, to include comments regarding the parameter-specific information gain.

 How do the phosgene retrieval diagnostics change for different atmospheric conditions (tropics, polar summer, polar winter) compared to mid-latitudes?

ANSWER:

Some differences exist. For example, in the FR polar summer scenario, the F11 interference error gives a larger contribution at high altitudes while the spectroscopic database error is smaller at low altitudes. Another example is that, in FR tropical conditions, the peak of F11 error is shifted towards higher altitudes with respect to the mid-latitude day example in figure 1. Further information can be found in the MIPAS systematic errors website maintained at the Oxford University (http://eodg.atm.ox.ac.uk/MIPAS/err/). Additional studies regarding the variability of the retrieval error with latitude and season were conducted during the characterization activities of Level 2 v.8 products. The Level 2 v.8 readme file (see fig. 4-121, page 154 of https://earth.esa.int/eogateway/documents/20142/37627/README_V8_issue_1.0_20201221.pdf ) shows that the relative random error components due to NESR and pT error propagation actually change their value with latitude and season. The changes, however, are mainly due to the variation of the average VMR profiles in the different latitudinal / seasonal scenarios, while the absolute errors are rather constant.

In the revised text, we added a sentence citing these additional analysis that includes different scenarios. Regarding fig. 2, the showed Averaging kernels are "typical" because they change really marginally with measurement scenarios.

Fig. 4: At the 50 hPa level, a significant bias/offset seems to be present between the FR and OR measurements. Can you provide an explanation for this offset? Most likely, it is due to the different retrieval characteristics of the FR and OR mode?

ANSWER: This is a well-known offset present in most of the MIPAS products. We know that it exists and we take it into account in the trend computation. It is mainly due to the different Micro-Windows (MWs) used for the retrieval in the FR and OR periods. Minor contributions are given also by the different vertical resolution and measurement vertical step in the two mission periods. In the revised text we added a sentence explaining this.

l207-209: It would be good to mention the total number of matches/profiles that have been available for comparison.

ANSWER: We added this information.

l210-212: It is pointed out that the MIPAS and ACE-FTS vertical profiles have been interpolated to the same levels to calculate their differences. However, how did you deal with the different vertical resolution of the data sets? Presumably, the vertical resolution of the MIPAS phosgene retrieval is different from the ACE-FTS data? Did you consider that systematic biases will arise in the comparisons due to the different vertical resolution of the data?

ANSWER: We are aware of the possible systematic biases due to the different vertical resolutions of the measurements. However, we do not have and we could not find ACE Averaging Kernels. The only information we found is that the ACE vertical resolution is about 3 km. If this is a really constant value, we can say that the vertical resolution has a minor effect below 15 km because MIPAS vertical resolution is about 3.5 km, very close to the ACE's one. On the other hand, MIPAS

vertical resolution starts to degrade at higher altitudes, reaching a value of 6 km at 20 km. The lower vertical resolution of MIPAS could therefore be responsible for a negative bias above the altitude of 15 km, mainly where the COCl2 peak is located.

We plan to include additional comments on this regard in the revised paper.

l234-238: Could the different vertical resolution of the data sets as represented by the averaging kernel also play a role in this comparison?

ANSWER: As we said in the previous answer, the contribution coming from different vertical resolutions between MIPAS and ACE is expected to be not negligible above the height of 15 km. As explanation of this point, we added a sentence in the text.

l265-273: This section looking at the comparison of the satellite data and the balloon data is also lacking some discussion regarding the (potentially) different vertical resolution of the data sets.

ANSWER: COCl2 MIPAS-balloon has a vertical resolution between 3 and 4 km, which is insofar consistent with MIPAS-ENVISAT. In the text, we indicated the MIPAS-balloon vertical resolution.

l336: In the conclusions section, it would be nice to include a few sentences about the broader implications of the study. Since MIPAS is out of order for about ten years, are other measurements being available or becoming available sometime soon to continue atmospheric phosgene measurements? Are the MIPAS phosgene measurements particularly important for specific applications in future work, e.g., evaluation of chemistry transport models?

ANSWER: We added two sentences at the end of the paper, explaining that these MIPAS measurements can be important to improve the chemical transport models and to understand the atmospheric sources and sinks of Cl-containing species. This study also provides an independent confirmation, obtained with a better temporal and spatial coverage, of the ACE results. In the future, the continuity of this study can be provided by ACE-FTS measurements, which are still being acquired.

Technical Corrections

l11-25: Merge these five paragraphs of 1-2 sentences each into just one?

l85: "lies" -> "flied" or "operated"

Table 1: apply AMT/Copernicus table format

l187: "polar nights" -> "polar winter" (?)

Fig. 8: the plots are quite small

Technical corrections were implemented in the revised paper.

---

## Author Comment (AC2)

General Comments

This is a reasonably short paper, presenting phosgene retrievals from the MIPAS ESA version 8 processor. The manuscript presents the global distribution, trends and comparisons with ACE-FTS data. Overall, I would say this is an adequate study, without being particularly ground breaking. AMT seems a good match for publishing this manuscript, although firstly there are a number of points that need to be addressed.

There have been a number of previous studies utilising satellite-derived phosgene (mainly from the ACE-FTS). It isn't clear to me what the motivation is for this study, so the authors should explain in more detail what this study tells us that we didn't already know.

ANSWER: Thank you very much for your comments that, we believe, will contribute to improve our paper.

Specific Comments

After reading this manuscript I am left with a number of questions that aren't addressed.

The manuscript glosses over the derivation of the a priori, which comes from the so-called IG2 profiles. For phosgene, these are averages of ACE v3.5/3.6 data. However, as with all satellite datasets, these ACE measurements (and therefore the a priori) are subject to bias. In fact, the ACE retrievals use completely different phosgene spectroscopy, and are likely not consistent with MIPAS spectroscopy.

Additionally, using ACE data as the a priori also makes any MIPAS-ACE comparison appear rather circular.

I would like to see a more in depth discussion of the choice of a priori, and its contribution to the retrieved MIPAS mixing ratios at each level. How does this contribution change over the profile, if at all?

ANSWER: Actually, COCl2 profiles included in the IG2 database stem from ACE data complemented with model data (in turn, the model has been tuned by comparisons with ACE data). While the IG2 database contains climatological profiles for 6 different latitude bands and 4 seasons, as a-priori we use a single profile, computed as a seasonal and latitudinal average of the IG2 tabulated profiles. In any individual retrieval this profile is then interpolated to the pressures corresponding to the altitudes of the vertical retrieval grid. Therefore, it is true that MIPAS v.8 products are not fully independent from ACE products, however we do not believe that such a weak dependency can invalidate the comparison with ACE measurements. The contribution of the a priori is different for each profile retrieval grid point and to quantify this contribution, we added to figure 2 a curve representing the information gain at each retrieval altitude. In the revised version of the paper the reader can evaluate the influence of the a-priori information and see that it becomes important at high altitudes, where the information gain decreases.

How does the new retrieval compare with the previous one in the study of Valeri et al? I understand there are some spectroscopy differences which should be considered here.

ANSWER: In the following plots we can see both COCl2 profiles of Valeri et al. (dotted lines) and ORM v.8 COCl2 profiles used for the study presented in this paper (solid lines). Plotted profiles are averages obtained exactly as in Valeri et al. (same days of data, same latitudinal bands and same seasons). The main differences occur below 100 hPa in polar and mid-latitude regions where negative differences between MIPAS v8 data and Valeri's profiles are visible. We used the same COCl2 spectroscopic database that is reported in Valeri article. However, Valeri et al. performed their analysis using version 7 of MIPAS level 1 data. That have a different correction of the time dependent nonlinearity of the detectors (see Kleinert et al, 2019). This and the different Micro-Windows (MWs) used for the retrievals and the different retrieval strategy can be responsible for the observed differences. Indeed, Valeri, differently from us, performed a simultaneous retrieval of COCl2 and CFC-11 because of their spectral interference occurring in his MWs.

[Figure]

3. Figure 9 indicates to me that the retrieval is far from perfect. There is a large offset/bias between OR and FR, which unfortunately casts doubt on the quality of the phosgene retrievals. What causes this? Is it linked to differences in the vertical resolution between OR and FR?

ANSWER: This offset is a well-known problem, present in most of the MIPAS products. We know that it exists, and we take it into account in the trend computation. It is mainly due to the different Micro-Windows (MWs) used for the retrieval in the FR and OR periods. Minor contributions are also due to the different vertical resolution and vertical sampling in the two mission periods.

4. There is no detailed discussion of the difference in spectroscopy between MIPAS and ACE. Figure 7 provides a plot of differences for just MIPAS. However, differences can also arise from the use of partition functions, which I suspect are handled differently for the older ACE linelist used in v3.5/3.6.

ANSWER: The difference in spectroscopy was already discussed in Valeri et al. Concerning the different partition functions, we have asked the ACE team for the partition function used in their analyses. In the plot below, we have plotted the correction that was applied to the line strength (the ratio between the reference partition function at 296 K and the partition function at a given temperature) as a function of temperature. The green line represents the one used for MIPAS and

the purple one for ACE. The plot shows that the differences are almost negligible, especially in the temperature range where phosgene contribution is maximum and, as a consequence, they don't influence results presented in the paper.

[Figure]

5. Comparisons of MIPAS trends with ACE-FTS are rather qualitative. For example, there is no attempt to recalculate ACE trends over the same time interval as the MIPAS data. The ACE trend is simply taken from a previous publication.

ANSWER: The main purpose of this paper is to estimate the COCl2 trend from MIPAS measurements. We compare our results to data found in literature, as it is commonly done and accepted.

6. How good is the retrieved MIPAS pressure and temperature and have these been validated? These are crucial in producing good quality phosgene trends.

ANSWER: Yes, they have been validated and all information about it can be found in the Product Quality Readme File (https://earth.esa.int/eogateway/documents/20142/37627/README_V8_issue_1.1_20210916.pdf). For the altitude scale, the bias found in the comparison against radiosonde is less than 20 m at low altitudes, increasing to a maximum of 100 m at an altitude of 10 hPa. No significant latitudinal dependencies are present. MIPAS temperatures are systematically colder than radiosonde and lidar data. The bias is less than 1 K in the stratosphere and it becomes about 2 K in the lower mesosphere. Also in this case, no significant latitudinal dependencies are present. Moreover, comparisons with radiosonde and lidar data show also a positive drift of about 0.4 K/decade in the differences between MIPAS and correlative temperature measurements. Note that such a drift in the MIPAS retrieved temperatures can't be directly translated into an error in the VMR trend. The temperature itself, which is retrieved from the same set of measurements, may actually compensate some instrumental / calibration drift which would otherwise affect the retrieved VMR.

Technical corrections:

line 31: photolysis

line 85: Fourier

line 335: resemble

Technical corrections were implemented in the revised paper.